# Experimental Investigation on a Bladed Disk with Traveling Wave Excitation

**DOI:** 10.3390/s21123966

**Published:** 2021-06-08

**Authors:** Luigi Carassale, Elena Rizzetto

**Affiliations:** Department of Mechanical Engineering, University of Genova, Via Opera Pia 15a, 16145 Genova, Italy; elena.rizzetto@edu.unige.it

**Keywords:** bladed disk, traveling wave, system identification

## Abstract

Bladed disks are key components of turbomachines and their dynamic behavior is strongly conditioned by their small accidental lack of symmetry referred to as blade mistuning. The experimental identification of mistuned disks is complicated due to several reasons related both to measurement and data processing issues. This paper describes the realization of a test rig designed to investigate the behavior of mistuned disks and develop or validate data processing techniques for system identification. To simplify experiments, using the opposite than in the real situation, the disk is fixed, while the excitation is rotating. The response measured during an experiment carried out in the resonance-crossing condition is used to compare three alternative techniques to estimate the frequency-response function of the disk.

## 1. Introduction

Bladed disks are key components of turbomachines, are highly stressed due to mechanical and thermal actions and are exposed to large-amplitude vibrations caused by several excitation sources including rotor-stator interaction and aeroelastic phenomena (e.g., [1,2]).

One peculiarity of these structures is their cyclic symmetry, i.e., the disk is ideally composed by a set of sectors (usually one per blade) that are nominally identical. The cyclic symmetry induces strong constraints on the modal properties of the disk. For a disk with *b* blades, the vibration modes are divided in families containing *b* modes. As a single sector is concerned, all the modes of the family have a similar shape (e.g., bending, torsional, etc.); however, considering the whole disk, the vibration amplitude of the blades is distributed circumferentially according to harmonic functions that can be characterized in terms of harmonic index (HI) or number of nodal diameters. For each HI, there exists a pair of modes sharing the same eigenvalues, with the exception of HI 0 and *b*/2 (if *b* is even) having single modes [3].

In practice, a small lack of symmetry appears to be due to manufacturing tolerances and ware, giving rise to a condition that is referred to as mistuning. Disks having a small mistuning retain the harmonic structure of their vibration mode shapes, but modes with the same HI have slightly different eigenvalues [4]. The most relevant consequence of mistuning is a significant increase in the maximum dynamic response during resonance crossing (e.g., [5]).

One of the most important excitation mechanisms for rotor blades is generated by the flow disturbances produced by the wakes and the potential-flow effects of stator blades located upstream and downstream (e.g., [1,2]). Due to the inherent periodicity of the angular coordinate, the circumferential distribution of the flow perturbations can be regarded as a sum of force components having harmonic circumferential shape. If observed in a rotating reference system attached to the rotor disk, these forces appear as traveling waves whose propagation velocity is given by the rotor speed. Similar to the mode shapes, each traveling-wave force component is characterized by its HI, which is referred to as engine order (EO).

The prediction of mistuning at design or construction stage is complicated due to its intrinsic randomness [6,7,8,9,10]. Mistuning can be identified experimentally by observing its effects on the dynamic response of the disk [11,12,13,14,15]. This activity is complicated for several reasons involving both experimentation and data processing. Limiting on the data processing side, difficulties arise from: (1) the excitation that is uncontrolled and unmeasured; (2) the signal-to-noise ratio that is favorable only in a small neighborhood of the resonance crossing; (3) resonances that are crossed quickly to prevent structural damages; (4) tests that are hardly repeatable and only one record is usually available.

The present paper describes the development of a test rig that is able to reproduce the main features of the problem described above in a controllable experimental environment. To avoid the difficulties involved in measuring rotating components, the disk is fixed, while the force is rotating. The bladed disk is simulated by a steel plate shaped on purpose, while the traveling-wave excitation is generated by a set of electromagnets.

Section 2 describes the test rig and its main components. Section 3 introduces a model to describe the force produced by the magnets. Section 4 describes the tuning of the system to obtain a disk with quasi-symmetric behavior. Section 5 presents the structural identification of the disk carried out through a standard input-output method and discusses some features of the disk response. Section 6 describes three methods for the identification of the disk excited by an unmeasured traveling-wave force crossing a resonance. Section 7 summarizes the results and provides some closing remarks.

## 2. Experimental Setup

The experimental setup is comprised of a still plate simulating an ideal bladed disk, a series of electromagnets, a set of amplifiers, sensors to measure the disk response and assess the applied forces, and a multi-channel signal generator. Figure 1a shows the conceptual schema of the setup. The setup is inspired by the experimental work documented by Bonhage et al. [16], but in the present case, the excitation source is produced by electromagnets, likewise in the test rig described by Firrone and Berruti [17].

### 2.1. Bladed Disk

The bladed disk was realized by a laser-cut carbon steel plate having *b* = 8 blades (Figure 1b). The plate thickness was 2 mm and its outer diameter was 450 mm. The disk was constrained at its center to a steel hub with a diameter of 60 mm through a bolted connection. To reduce the effect of external vibrations the whole setup was supported by an optical table with air-bearing insulation devices. From preliminary numeric analyses we observed that the first mode family of the disk (bending of the blades) was well-separated from the others and the natural frequencies are close to 30 Hz. It was also observed that the behavior of neighbor blades is strongly coupled.

### 2.2. Magnets

The excitation force was produced by electromagnets realized through a coil wound around a ferrite core (Figure 2a). The length ℓ_0_ of the air gap between the magnet core and the disk was adjustable due to the spring-mounted support shown in Figure 2b. In its nominal position ℓ_0_ = 5 mm. Figure 3 shows the impedance *Z* of the electric circuit. For reasons that will be explained afterward, the working frequency of the circuit during the experiments was in the neighborhood of 15 Hz where |*Z*| as close to 4 Ω and its phase angle was relatively small.

### 2.3. Amplifiers

The eight magnets were driven by eight linear power amplifiers. They were class-B units, built following a push-pull configuration, with 200 W nominal power. Figure 4 shows the voltage gain *G* of the amplifiers. In the frequency range used for the experiments described herein, the modulus of the gain was about 35 and the phase angle was about 15°.

### 2.4. Instruments for Measurement

The response of the disk was measured by eight accelerometers, type PCB-333B30 (PCB PIEZOTRONICS, Depew, NY, USA) with a sensitivity of 100 mV/g mounted near the blade tips. A careful cable routing was recognized as important to ensure the repeatability of the experiment. The final cable layout is shown in Figure 1b.

The magnet force was not measured but was indirectly estimated from the measurement of the current in the magnets through current transformers type LEM ATO-10-B333-D10 (Geneve, Switzerland).

The analog signals were acquired through an NI system equipped with PXIe-4492, NI (Austin, TX, USA) boards. The sampling frequency was set to 2000 Hz.

### 2.5. Signal Generation

The input signals provided to the eight amplifiers are generated digitally by an NI PXIe-6738 board. In all the experiments described herein, the excitation was constituted by chirp functions in the form:(1)$$v_{k}\left(t\right)=V\cos\left(\omega_{0}t+\frac{1}{2}\dot{\omega }t^{2}+\alpha_{k}\right)\;(k=0,\ldots ,b-1)$$
where *k* is the blade count, *V* is the signal amplitude, *ω*_0_ is the starting circular frequency, and $\dot{\omega }$ is its derivative with respect to time *t* and phase angle α*_k_*. According to Equation (1), the eight amplifiers receive the same input signal with a possibly different phase angle.

## 3. Magnetic Force

The *k*th electromagnet and the neighboring blade form a magnetic circuit in which the magnetic flux travels partially within the ferrite core and partially across the air gap. According to Hopkinson’s law, the magnetic flux density for the *k*th magnet is given by the expression:(2)$$B_{k}=\frac{Ni_{k}}{\frac{\ell_{c}}{\mu }+\frac{\ell_{k}}{\mu_{0}}}$$
where *N* is the turns in the winding, *i_k_* is the current in the magnet, ℓ*_c_* is the length of the magnetic circuit in the ferrite core, ℓ*_k_* is the width of the air gap, and µ*_c_* and µ_0_ are the magnetic permeabilities of the core and the air, respectively. The width of the air gap changes with time during the blade vibration, i.e., ℓ*_k_*(*t*) = ℓ_0_ + *u_k_*(*t*), where ℓ_0_ is the gap width in static condition and *u_k_*(*t*) is the displacement of the *k*th blade. However, assuming $u_{k}\ll \ell_{0}$, such a dependency is disregarded. Additionally, since $\mu_{c}\gg \mu_{0}$, the expression for the magnetic flux simplifies as
(3)$$B_{k}=\frac{\mu_{0}Ni_{k}}{\ell_{0}}$$

The force exerted by the electromagnet on the *k*th blade is given by the expression:(4)$$\begin{array}{ll} p_{k} & =\frac{B_{k}^{2}S}{2\mu_{0}}\\ & =\frac{\mu_{0}N^{2}S}{2\ell_{0}^{2}}i_{k}^{2}\\ & =Pi_{k}^{2} \end{array}$$
where *S* is the cross-section area of the magnet core. Equation (4) states that the forces *p_k_* are proportional to the square of the currents *i_k_* in the magnets through the constant *P*. In principle such a constant may be obtained theoretically; however, due to the leakage flux, its experimental identification appears more appropriate. In the nominal setup it was estimated as *P* = 1.65 *N*/*A*^2^.

The relationship between the magnetic force *p_k_* and the excitation signal *v_k_* can be obtained by substituting Equation (1) into Equation (4) and assuming that the amplifier gain and magnet impedance are constant within the frequency band of interest. It yields:(5)$$p_{k}\left(t\right)=\frac{P}{2}{\left(\frac{V\left|G\right|}{\left|Z\right|}\right)}^{2}\left[1+\cos\left(2\omega_{0}t+\dot{\omega }t^{2}+2{\alpha^{\prime }}_{k}\right)\right]$$
where ${\alpha^{\prime }}_{k}=\alpha_{k}+∠G-∠Z$ is the phase of the current in the magnets. It can be observed that the force is composed by a constant term plus a harmonic component having a frequency twice that of the input signal. Additionally, since the amplifiers and magnets are the same for every blade, the phase lag of the force acting on adjacent blades is twice the phase lag of the corresponding input signals. Accordingly, a traveling wave excitation with EO *h* is produced by setting in Equation (5)
(6)$$\alpha_{k}=\pi \frac{kh}{b}\;\left(\begin{array}{l} k=0,\ldots ,b-1\\ h=0,\ldots ,\frac{b}{2} \end{array}\right)$$

## 4. Tuning of the Experimental Setup

The test rig was intended to reproduce a nearly-symmetric bladed disk, as this condition implies high modal density and is challenging for system identification algorithms. Unfortunately, even if the disk was realized using an accurate laser-cut procedure, its dynamic behavior appeared initially slightly symmetric. To investigate this issue, the dynamic behavior of each individual blade was isolated by adding masses to all the blades but the measured one. This procedure created a wide separation between the vibration mode involving the measured blade and the other vibration modes of the same family, whose natural frequencies are shifted downward [18]. The isolation of the blades was judged successful as the ratio between the receptance of the *b*−1 blades with the attached mass and the receptance of the isolated blade was always lower that 4% with an average value about 0.4%.

The system identification was carried out using the force estimated from the measured current as the input and the blade acceleration as the output. Figure 5a shows the modulus of the frequency response function (FRF) for the eight blades of the disk measured in the initial experimental setup. Blade-to-blade differences both in terms of natural frequency and amplitude are clearly visible.

The blade behavior was tuned by adding small masses at the blade tips (Figure 2b), to modify their natural frequencies. Additionally, the amplitude of the force acting on the blades was tuned by modifying the blade-to-magnet distance to equalize the maximum value of the FRFs. Figure 5b shows the FRF of the blades after the tuning procedure.

The tuning procedure described above also allowed us to verify the repeatability of the disk behavior. This aspect was relatively problematic at the beginning due to the support hub, as well as the sensor wiring. The problem was circumvented by an accurate redesign of the hub including a precise centering feature and a careful cable routing and fixture. After these modifications, no relevant repeatability issues were noted.

## 5. Dynamic Behavior of the Disk

The dynamic behavior of the disk was analyzed through an extensive set of measurements. The excitation was produced by multiple repetitions of a chirp-type force crossing the resonance, applied separately to each of the eight blades. The parameters used in Equation (1) were *V* = 50 mV, *ω*_0_ = 2π∙10 Hz, and $\dot{\omega }$ = 2π∙0.08 Hz/s to provide an excitation in the frequency range between 20 and 40 Hz. The response of each blade was measured in terms of acceleration ${\ddot{u}}_{k}(t)$ (*k* = 0, …, *b* − 1), while the force was observed, indirectly, measuring the current *i_k_*(*t*) in the (only) active magnet.

The repetition for all the blades of the described procedure enabled the estimation of the full (*b* by *b*) FRF matrix **H** of the system.

Figure 6 shows the amplitude (lower triangle) and phase (upper triangle) of the components of **H** evaluated adopting the *H*_1_ estimator [19] (solid blue line). Regarding the direct terms, the behavior of the disk appeared symmetric, while some lack of symmetry was visible in the cross-terms. In particular, the coupling terms of neighboring blades were repeated every second blade.

The experimental FRF **H** was fitted in the frequency range between 20 and 40 Hz by a 16th-order (twice the number of the blades *b*), continuous time state-space model:(7)$$\{\begin{array}{c} \dot{x}=Ax+Bp\\ \ddot{u}=Cx+Dp \end{array}$$
where $p={\left[p_{0}\ldots p_{b-1}\right]}^{T}$ is the input (force) vector, $\ddot{u}={\left[{\ddot{u}}_{0}\ldots {\ddot{u}}_{b-1}\right]}^{T}$ is the output (acceleration) vector and **x** is the state vector. The system matrices **A**, **B**, **C** and **D** were identified using a frequency-domain implementation of the subspace method [20]. The FRF of the model (7) was obtained from the systems matrices as
(8)$$H\left(\omega \right)=C{\left(j\omega -A\right)}^{-1}B+D$$
and is reported in Figure 6 by dashed lines. Notably, the model matches the experimental FRF very well.

The natural frequencies *ω_r_*, damping ratios ξ*_r_*, and mode shapes **ϕ***_r_* of the disk are obtained from the relationships:(9)ωr=|λr|ξr=−Re[λr]|λr|ϕr=Cψr
where *λ_r_* and **ψ***_r_* are, respectively, the eigenvalues and the eigenvectors of **A**. Table 1 shows the natural frequencies and damping ratios of the eight identified vibration modes, together with the HI of the corresponding mode shape. Figure 7 shows the mode shapes estimated from data (solid line) and the best fitting harmonic function (dashed line).

The vibration modes strictly followed the harmonic structure typical of cyclic symmetric systems with the mode pairs 1–2, 4–5, and 6–7 having, respectively, HI 1, 2, and 3. Modes 3 and 8 had HI 0 and 4, respectively. The frequency separation for mode pairs 1–2 and 6–7 was small; thus, the corresponding peaks of the FRF merged. On the contrary, the peaks of the mode pair 4–5 appeared separated due to the very low damping. The damping ratios were below 10^−3^ except for mode 3 having HI 0.

## 6. Identification from Resonance-Crossing Response at a Traveling-Wave Excitation

The bladed disk that was identified by the standard input-output procedure described in Section 5 was analyzed based only on the measurement of its transient response to a traveling wave excitation crossing the resonance. The excitation was not measured, but it was assumed that its instantaneous frequency is known (in practical applications involving rotating machinery, this information is obtained from the tachometric probe). Additionally, a single resonance crossing was assumed to be available, as in practical conditions, run-up or run-down operations can often be measured only once.

In the case of traveling wave excitation, the force vector can be completely expressed on the basis of the force acting on blade 0:(10)$$p\left(t\right)=\left[\begin{array}{c} 1\\ e^{j2\pi \frac{h}{b}}\\ \vdots \\ e^{j2\pi \frac{h(b-1)}{b}} \end{array}\right]p_{0}\;\left(t\right)=e_{h}p_{0}\left(t\right)$$

In this condition, the state-space model (7) can be rewritten as:(11)$$\{\begin{array}{c} \dot{x}=Ax+B_{h}p_{0}\\ \ddot{u}=Cx+D_{h}p_{0} \end{array}$$
where **B***_h_* = **Be***_h_* and **D***_h_* = **De***_h_*. The corresponding FRF can be obtained from Equation (8) and reads
(12)$$H_{h}\left(\omega \right)=H\left(\omega \right)e_{h}$$

The state-space Models (7) and (11) share the same modal properties, as these are determined by matrices **A** and **C**. However, while the identification of model (7) requires a full input-output test, Model (11) can be identified through the knowledge of the FRF (12), which can be estimated from the response to a traveling wave excitation (provided that the model (11) is controllable). 

The FRF **H***_h_*(*ω*) can be estimated from the transient response following different strategies. Three alternatives of increasing complexity and computational cost are considered.

### 6.1. Demodulation

The simplest method to obtain an unscaled estimation of the FRF is probably based on the demodulation of the measured response. The response of each blade is assumed as an amplitude-phase modulated signal:(13)$${\ddot{u}}_{k}\left(t\right)=U_{k}\left(t\right)\cos\phi_{k}\left(t\right)\;\left(k=0,\ldots ,b-1\right)$$
where the amplitude *U_k_*(*t*) and the phase ϕ*_k_*(*t*) are slowly-varying functions of time. These functions can be evaluated according to different methods, including order tracking algorithms that can be implemented in online processing tools. In the present case, the demodulation of the measured signal was computed off-line by the Hilbert transform.

Since the frequency of the magnetic force (twice the frequency of the driving signals) is known as a function of time, the response amplitude and phase may be regarded as functions of *ω* and considered as an estimate of the FRF (unscaled as the force is unmeasured).
(14)$$H_{h}^{\left(D\right)}\left(\omega \right)=\left[\begin{array}{c} U_{0}\left(\omega \right)e^{j\phi_{0}\left(\omega \right)}\\ \vdots \\ U_{b-1}\left(\omega \right)e^{j\phi_{b-1}\left(\omega \right)} \end{array}\right]$$

This assumption is correct if the system resonance is crossed very slowly and the transient effects are negligible. For a perfectly tuned disk or a disk with an isolated pair of mistuned modes, the mentioned condition is governed by the reduced sweep rate [21]:(15)$$\kappa =\frac{\dot{\omega }}{\omega_{n}^{2}\xi^{2}}$$
where $\dot{\omega }$ is the time derivative of the force frequency; *ω_n_* and ξ are, respectively, the natural frequency and damping ratio of the active modes. It is demonstrated by an asymptotic analysis that transient effects are negligible if κ≪1 [21].

### 6.2. Fourier Transform with Simulated Input

Since the instantaneous frequency of the force is known, an idealized expression for the force acting on blade 0 can be written as:(16)$$p_{0}\left(t\right)=\cos\phi \left(t\right)$$
where
(17)$$\phi \left(t\right)=\int_{0}^{t}\omega \left(\tau \right)d\tau$$

Accordingly, the FRF can be estimated as
(18)$$H_{h}^{\left(FT\right)}\left(\omega \right)=\frac{{ℱ}_{\omega }\left[\ddot{u}\right]}{{ℱ}_{\omega }\left[p_{0}\right]}$$
where ${ℱ}_{\omega }$ represents the Fourier transform calculated at the frequency *ω*:(19)$${ℱ}_{\omega }\left[x\right]=\int_{-\infty }^{\infty }x\left(t\right)e^{-j\omega t}dt$$

In principle, this method is correct if the force amplitude and phase, as well as the mechanical parameters of the system do not change significantly during the resonance crossing. A major drawback is its weak resistance to noise as the availability of a single transient signal prevents the possibility of averaging.

### 6.3. Wavelet Transform along the Excitation Line

The continuous wavelet transform of a square-integrable function *x*(*t*) is defined as (e.g., [22])
(20)$$W_{s,\tau }\left[x\right]=\int_{-\infty }^{\infty }x\left(t\right)\psi_{s,\tau }^{*}\left(t\right)dt$$
where the superscript * indicates the conjugate and the two parameters *s* and τ represent, respectively, the central time and the scale of the wavelet. The basis functions $\psi_{s,\tau }$ are obtained by translation and dilation of a mother wavelet as:(21)$$\psi_{s,\tau }\left(t\right)=\frac{1}{\sqrt{s}}\psi \left(\frac{t-\tau }{s}\right)$$

For the choice of the mother wavelet, several alternatives are available. In the present study, the Gabor wavelet is adopted [22]:(22)$$\psi \left(t\right)=\sqrt[{4}]{\frac{2\sigma_{\omega }^{2}}{\pi }}e^{j\omega_{c}t}e^{-\sigma_{\omega }^{2}t^{2}}$$
where *ω_c_* is the central frequency and σ_ω_ is the frequency spread.

Using the wavelet transform, the unscaled FRF **H***_h_* can be estimated as [23]:(23)$$H_{h}^{\left(WT\right)}\left(\omega \right)=W_{s,\tau }\left[\ddot{u}\right]e^{-j\phi \left(t\right)}\;\left(s=\frac{\omega_{c}}{\omega \left(t\right)},\;\tau =t\right)$$
where the scale and the central time are selected to follow the trajectory of the excitation on the time–frequency plane.

In principle, the described procedure is independent of the choice of the wavelet family that is employed, and the use of the Gabor wavelet may be questionable as it is neither rigorously analytic nor compatible [22]. However, in Carassale et al. [23], it was observed that, for this kind of application, these theoretical drawbacks are not relevant. However, the advantage of Gabor wavelet of having the minimum possible time-frequency spread may be a positive feature.

### 6.4. Application

The three techniques described above were applied to estimate the FRF from one record measuring a resonance crossing of a traveling wave force with EO 1 and frequency increasing from 30 Hz to 36 Hz in 120 s. The reduced sweep rate (related to vibration modes 1 and 2) was κ = 16.

Figure 8 shows the response measured on blade 0 together with the spectrogram obtained from the wavelet transform computed assuming σ_*ω*_/*ω_c_* = 10^−3^. The resonance amplification is clearly visible at about *t* = 60 s. After that time, the amplitude of the response is strongly modulated due to the beating between the forced response and the free-decay transient response. The observation of the spectrogram provides a deeper description of the phenomenon. It can be noted that:The inclined straight line crossing the spectrogram is the direct effect of the excitation and can be used to estimate its instantaneous frequency if it is not directly measured. The wavelet transform of Equation (23) is computed along this line.Even if the maximum amplification of the response appears to be crossing the resonance of the modes with HI 1, all the other modes are excited as well.The response about 33 Hz on the left extremity of the spectrogram is due to the transient related to the start of the experiment; this effect (not relevant for the present purpose) is almost invisible in the time-domain data, but can be clearly observed in the spectrogram.

Figure 9 shows the FRF of the disk excited by a traveling wave of EO 1, estimated by demodulation, Fourier transform, and wavelet transform. The target FRF (blue line) is obtained through Equation (12). The results are pertinent for blade 0; however, the plots for the other blade are analogous. The FRF is dominated by the peak due to modes 1 and 2 (the ones having the same HI of the excitation), but also contains much lower peaks corresponding to the mode pairs 4–5, 6–7, and the isolated mode 8. These latter modes are excited due to the lack of symmetry of the disk or the imperfect realization of the traveling wave excitation. Their response is one order of magnitude lower than the response of the mode pair 1–2, but, at least in principle, can be used for system identification purpose. All the identification methods presented are able to detect the peak and the phase shift related to mode 1–2; however, the resonance peak identified by demodulation is lower than the real one, slightly shifted rightward, and thicker than expected. All these aberrations are widely documented effects of transience [21]. Modes 4 and 5 (HI 2) are visible both in the FRF estimated by Fourier transform and wavelet transform. Modes 6 and 7 (HI 3) and 8 (HI 4) are visible only in the FRF obtained by the wavelet transform.

## 7. Conclusions

The realization of the text rig, the execution of the experiments and the analysis of the acquired data suggest the following closing remarks:

A careful tuning of the experimental setup led to nearly-symmetric condition that can be used to test system identification algorithms. The vibration modes of the disk have harmonic shape and the natural frequencies of modes with the same HI have a separation lower than 1%. Despite this condition, the effect of mistuning is still important enough to enable the excitation of all of the modes on the disk by a traveling-wave excitation of EO 1.

A series of magnets driven by suitable signals can be used to produce a traveling-wave force. This excitation was applied to a fixed bladed disk, but, in principle, it can be used to excite a rotating disk at a non-synchronous frequency.

The estimation of the (unscaled) FRF from a resonance-crossing test was successful; however, the quality of the results depends on the technique that is adopted. Among the three considered approaches, the wavelet transform calculated along the excitation line provided the most accurate results, being able to also detect the modes that are weakly excited due to experimental imperfections.

The system identification problem is challenging mostly due to the high modal density. This feature tends to be emphasized by increasing the number of blades composing the disk. Further experiments should be planned to consider this aspect, as well as the case in which only a partial set of blades are observed.

## Figures and Tables

**Figure 1 sensors-21-03966-f001:**
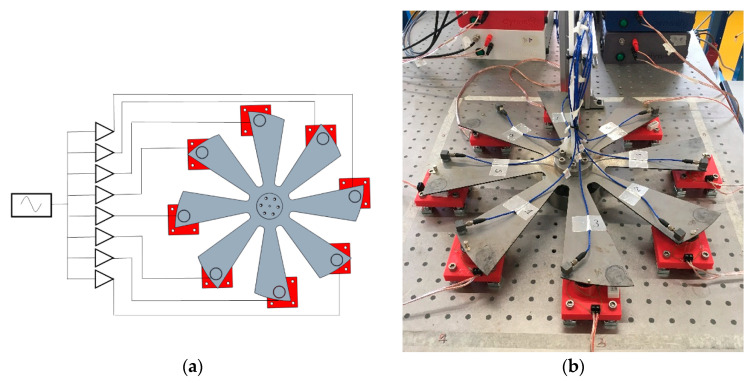
Experimental setup: (**a**) conceptual schema; (**b**) bladed disk.

**Figure 2 sensors-21-03966-f002:**
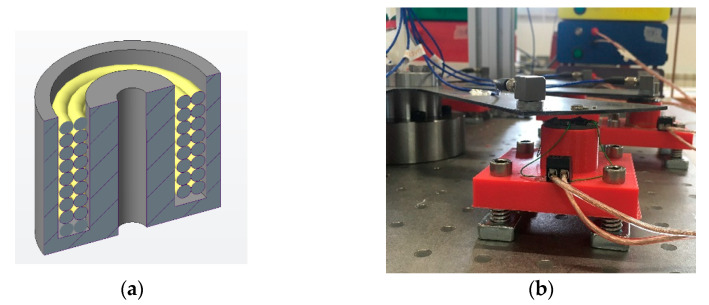
Excitation devices: (**a**) cross section of the magnet; (**b**) spring-mounted support.

**Figure 3 sensors-21-03966-f003:**
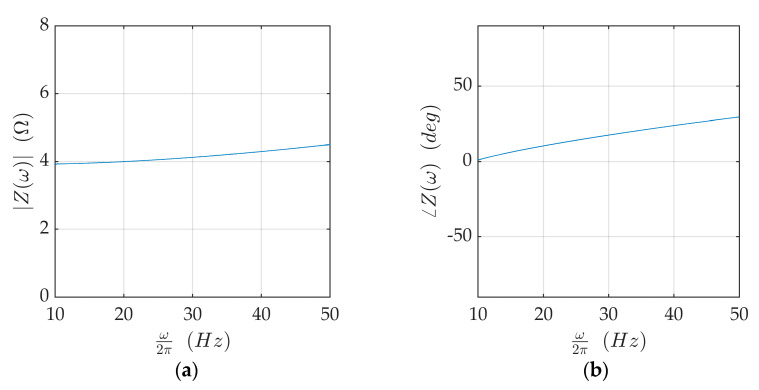
Impedance of the electric circuit: (**a**) amplitude; (**b**) phase angle.

**Figure 4 sensors-21-03966-f004:**
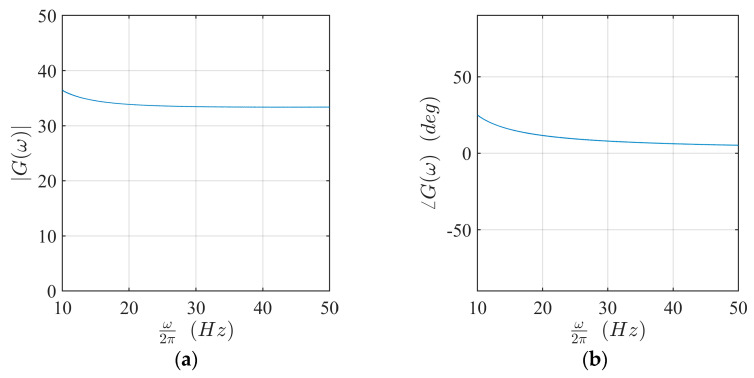
Voltage gain of the amplifiers: (**a**) amplitude; (**b**) phase angle.

**Figure 5 sensors-21-03966-f005:**
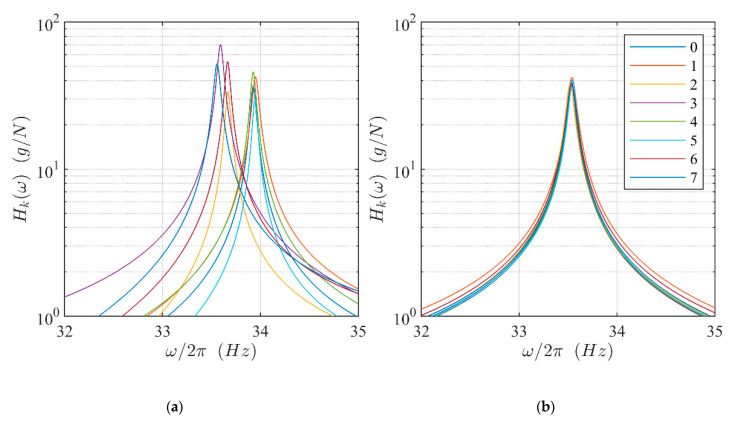
FRF of the isolated blades: (**a**) before tuning; (**b**) after tuning.

**Figure 6 sensors-21-03966-f006:**
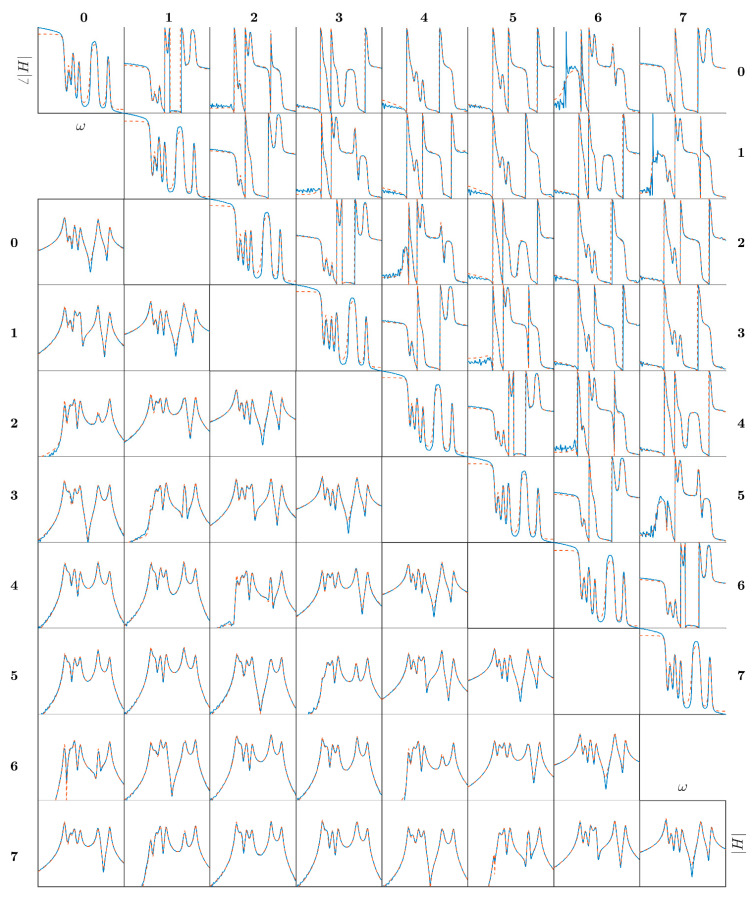
FRF of the bladed disk. Modulus lower triangle; phase upper triangle; frequency range, 32–35 Hz.

**Figure 7 sensors-21-03966-f007:**
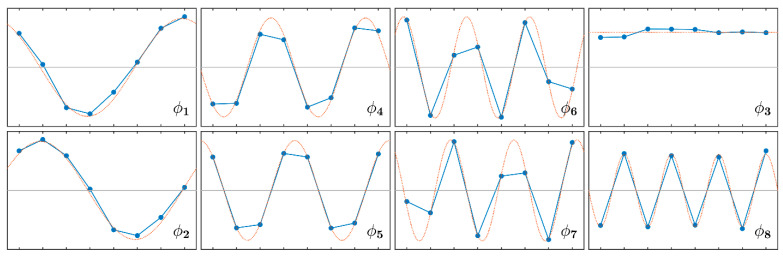
Mode shapes of the disk: identified (solid line); best fitting harmonics (dashed lines).

**Figure 8 sensors-21-03966-f008:**
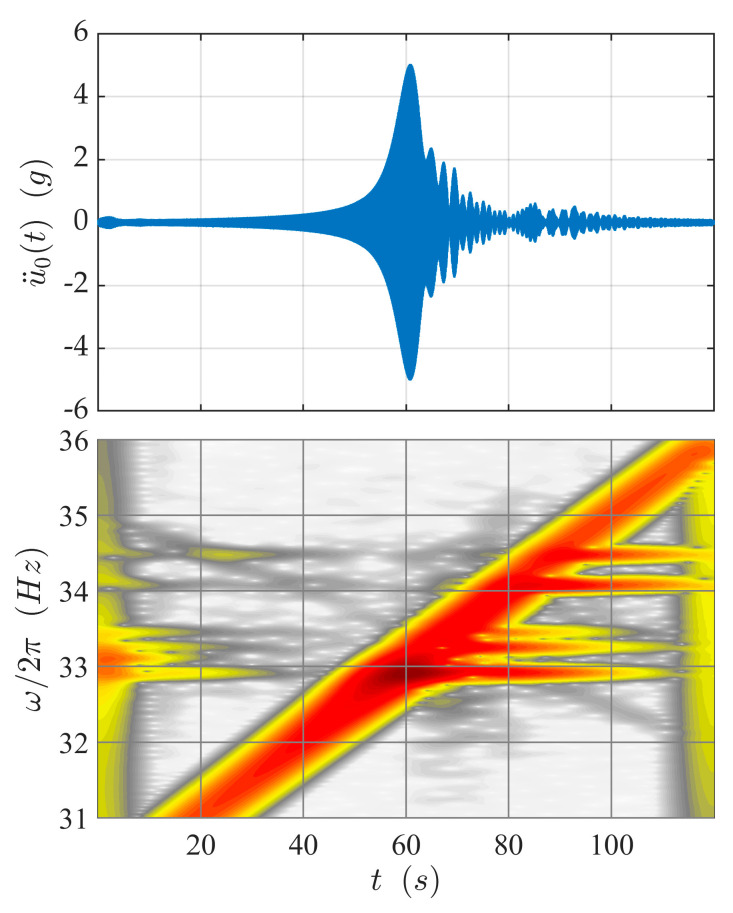
Response of blade 0 during the resonance crossing of a traveling-wave force with HI 1.

**Figure 9 sensors-21-03966-f009:**
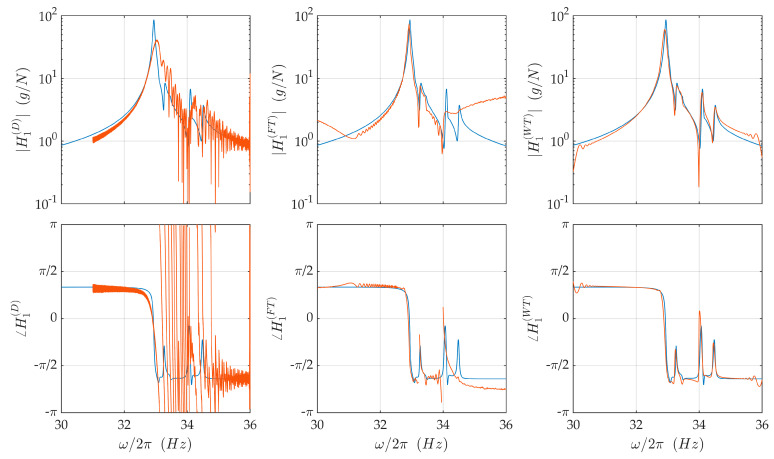
FRF of the disk excited with a traveling wave of HI 1. Estimation by demodulation, Fourier transform, and wavelet transform for blade 0.

**Table 1 sensors-21-03966-t001:** Modal properties of the disk.

Mode	HI	*ω_r_*/2π (Hz)	ξ*_r_* (×10^−3^)
1	1	32.92	0.70
2	1	32.94	0.76
3	0	33.09	1.15
4	2	33.28	0.78
5	2	33.47	0.81
6	3	34.10	0.71
7	3	34.13	0.76
8	4	34.50	0.72

## Data Availability

The data presented in this study are available on request from the corresponding author.

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
