# Peer review of "Experimental Investigation on a Bladed Disk with Traveling Wave Excitation"

_sensors, 2021, doi:10.3390/s21123966_

Round 1

Reviewer 1 Report

The paper show results of an experimental investigation of bladed disk dynamics.

The paper is well written, scientifically sound and of interest to the scientific community.

The main contribution is the comparison between different experimental techniques for FRF estimation. It would have been interesting to compare modal properties identified by FRF to those identified by single-point excitation.

I wrote my main comments and requests in the pdf paper.

Author Response

We wish to thank the reviewer very much for his careful reading of the paper. His comments contributed to significantly improve our paper. Besides, he recognized a gross mistake that we made in the final editing of a figure. Also for this, we wish to thank him very gratefully.

We addressed all his observations as indicated below (in red).

There are papers from University of Michigan and Turin Politecnico, where excitation is controlled and estimated. The authors should comment about these papers in the introduction…

The two references suggested by reviewer are fully relevant and we do regret not having considered them during the writing of our paper. In the new version of the paper, both the references are cited.

In my personal experience, getting isolated blade mode shapes from blisks is rather difficult. Have the authors checked the receptance of the N-1 blades with the attached mass? Was it negligible compared to the isolated blade's?

The ratio between the receptance of the N-1 blades with the attached mass and the receptance of the isolated blade was always lower that 4% with an average value about 0.4%. On this basis, we believe that the isolation of the blades was rather successful. We added a comment on this point.

Why modifying the blade-magnet distance contributes to tune the model? I would say that it only tunes the excitation, increasing the accuracy of the engine order type traveling excitation.

We totally agree with this comment (and it is what we meant).

In the new version of the paper, the tuning procedure is explained more clearly separating the role of the added mass (to tune the mechanical response) and the magnet distance (to tune the engine-order excitation).

Are you sure about the out-of-diagonal lower triangle results? With a stationary excitation, some mode shapes (those, whose nodal diameter passes through the excitation point) should not be excited (or at least poorly excited) in a well tuned assembly. Can you also comment on the number of peaks? In theory, assuming a cyclically symmetric system, they should be 5.

There was indeed a gross mistake in the lower part of Figure 5, which did not even correspond to our comments that referred, instead, to the correct. Indeed, the mistake originated in the final formatting of the figure. We are very grateful to the reviewer for having recognized this error. A wider explanation, including a comment of the peaks has also been added.

Can the authors comment about the multiple peaks in the FRF? Are they due to the non perfectly harmonic mode shapes or to the non perfectly engine-order type excitation?

A more accurate description of this feature and a comment have been added

Can the authors clarify in the paper why the bladed disk was modified trying to tune it? Wouldn't the experimental results be meaningful if the original bladed disk was used?

We tried to reproduce a quasi-symmetric condition as it is the most realistic one and it is the most challenging from the identification point of view. We added a comment to explain this motivation.

As mentioned by the authors in the paper, the blades are strongly coupled. As a result, the resonance peaks of the 1st family are distributed over a rather wide frequency range. In case of bladed disks with a larger number of blades, peaks might be not easy to distinguish. Could it be an issue for any of the identification techniques described in the paper?

This comment is definitely pertinent; however, we do not have a ready answer. It is possible that with a larger number of blades or a, in general, a higher modal density, the processing methods become less accurate. However, this issue would probably affect in a similar way all the methods that have been compared, without changing the validity of the conclusions.  A comment on this issue has been added.

Reviewer 2 Report

Dear Authors,

I have read your paper with pleasure and attention.

In my opinion, the manuscript Experimental investigation on a bladed disk with traveling wave excitation  presents original research and could be interesting for readers of the Sensor Journal. The motivation is clear. The object of study, as well as the results, are comprehensively described providing valuable conclusions.

The paper is organised in a logical manner. The state of art covers the main results in the field, including the authors’ own results. The contributions of the paper are clearly stated in the Introduction chapter. The application of the proposed method in laboratory tests was presented, the results were discussed and accurate conclusions were drawn.

I have no objections to publishing this paper. However, due to the listed below drawbacks, my recommendation is " Accept minor revision". In my opinion, several aspects require clarification. Please revise and add some comments and improvements according to the following:

-how repeatable are the measurement results obtained on the laboratory model shown and described in the Experimental setup chapter? Whether analyzes were performed to confirm the stability of the setup used?

- in the article there is information that wawelt analysis was used, with the use of Gabor wavelet. Have other wavelets been used and checked if it is possible to obtain better results?

Author Response

We wish to thank the Reviewer for his careful reading and his constructive and pertinent comments. We addressed all his observations as indicated below (in red).

-how repeatable are the measurement results obtained on the laboratory model shown and described in the Experimental setup chapter? Whether analyzes were performed to confirm the stability of the setup used?

Repeatability has been a serious issue (actually a nightmare) on the beginning of the experimentation. It was due to the support hub as well as the sensor wiring. The problem was circumvented buy an accurate redesign of the hub including an accurate centering feature and a careful cable routing and fixture. After these modifications, no relevant repeatability issues have been noted.

In the new version of the paper we added some details to explain this refinement process and to provide indications that may be useful to reproduce the experiment.

- in the article there is information that wavelet analysis was used, with the use of Gabor wavelet. Have other wavelets been used and checked if it is possible to obtain better results?

For this specific case, we did not test any other wavelet family as we started for the experience of other test campaigns regarding full-scale tests on turbines. In these previous tests (also described in a referred paper) we tested several wavelet families concluding that the theoretical disadvantages of the Gabor wavelet (it is not rigorously analytic nor compatible) are not relevant for the specific application, while its advantage (minimum time-frequency spread) may be a positive feature.

In the new version of the paper we add a note about the choice of the wavelet family.